# PCR-based detection and phylogenetic analysis of *Candidatus* Liberibacter asiaticus in citrus orchards across Nepal

Richa Giri[1,2], Bal Kumari Oliya (ID)[3]*, Siddartha Gautam[1,4], Krishna Das Manandhar[1]

**1** Central Department of Biotechnology, Tribhuvan University, Kirtipur, Kathmandu, Nepal, **2** Warm Temperate Horticulture Centre, Ministry of Agriculture and Livestock Development, Kirtipur, Kathmandu, Nepal, **3** Seed Quality Control Centre, Ministry of Agriculture and Livestock Development, Hariharbhawan, Lalitpur, Nepal, **4** Floriculture Development Centre, Godawari, Lalitpur, Nepal

* balkumarioliya1@gmail.com

## Abstract

Citrus greening disease, also known as huanglongbing (HLB), is caused by the gram-negative α-proteobacteria *Candidatus* Liberibacter species. This disease poses a significant threat to citrus production worldwide, including in Nepal. This study aimed to perform the diagnosis and phylogenetic analysis of the citrus greening pathogen in Nepal using both conventional PCR and computational methods. A total of 1,026 citrus leaf samples were collected from thirteen districts across six provinces in the country. PCR-based diagnosis was performed using the primer set Las606/LSS, which targets the 16S rRNA gene of *Candidatus* Liberibacter asiaticus. The reliability of the PCR was validated by including previously confirmed positive and negative controls in every run. Furthermore, each assay was performed in triplicate to ensure consistency and reproducibility of results. Additionally, 16S rRNA gene sequencing was performed using Sanger sequencing for five samples. The obtained sequences were deposited in GenBank and a phylogenetic tree was constructed based on these sequences. Among the 1,026 samples tested, 255 were positive, indicating the widespread distribution of HLB across Nepal. All consensus sequences from Nepal showed strong evolutionary relatedness within the *Ca.* L. asiaticus cluster, with over 99% genetic similarity to reference sequences from various parts of the world. Phylogenetic analysis revealed that the Nepalese sequences were closely related to *Ca.* L. asiaticus sequences from India (Punjab and Meerut) and sequences obtained from different regions of Nepal clustered closely together. The molecular findings from this study reveal a high prevalence of citrus greening disease across Nepal and underscore the urgent need for integrated management policies, including the use of certified clean planting material and vector (psyllid) control programs. The generated sequence data serves as a vital resource for developing regional diagnostic tools and guiding future surveillance strategies to mitigate HLB's impact on the world's citrus industry.

**Data availability statement:** All relevant data are within the manuscript and its Supporting Information files.

**Funding:** The author(s) received no specific funding for this work.

**Competing interests:** The authors have declared that no competing interests exist.

## Introduction

Citrus is grown in tropical and subtropical climates around the world, with both fresh and processed products remaining in high global demand [1]. Although its origin is generally considered to be in Southeast Asia [2], the major producers today are China, Brazil and the United States [3]. In Nepal, citrus species constitute a major fruit group due to the suitable agro-climatic conditions of the mid-hills [4] and account for 20.81% of the total fruit production, with cultivation widespread across the country [5]. Commonly cultivated species include *Citrus reticulata*, *C. sinensis*, *C. grandis*, *C. limon*, *C. aurantifolia* and *Poncirus trifoliata* [6]. Among these, Mandarin oranges (*C. reticulata*) account for approximately 63% of Nepal's citrus production, followed by sweet oranges (*C. sinensis*) at 16% and acid limes (*C. aurantifolia*) at 15% [4]. This dominance highlights the crop's key role in improving food security, nutrition and employment [7]. Despite this significant contribution, domestic production remains inadequate to meet the overall demand [8].

Among the factors contributing to citrus decline, citrus greening (huanglongbing, HLB), also referred to as yellow shoot disease, is regarded as the most severe cause of citrus decline globally [ 9–14] and poses a major threat to the citrus industry worldwide, including Nepal [6,15–19]. Gram-negative bacterium *Candidatus* Liberibacter spp. is the causative agent of citrus greening disease, which is primarily found within the phloem tissues of the plant [18,20]. The three primary species of *Candidatus* Liberibacter associated with citrus greening disease are *Candidatus* Liberibacter asiaticus (*Ca.* L. asiaticus), *Candidatus* Liberibacter africanus (*Ca.* L. africanus) and *Candidatus* Liberibacter americanus (*Ca.* L. americanus). *Ca.* L. asiaticus is commonly found in Asia, *Ca.* L. africanus is found in Africa, while *Ca.* L. americanus is found in South America [21,22]. There is a limitation in understanding the biology of this pathogen since there is no successful pure (axenic) culture, which limits the functional genomic analysis, leading to challenges in HLB disease management [23]. Despite several attempts being made to culture this pathogen, only short-term culture has been reported [24]. HLB is a vector-borne disease transmitted by psyllids, also known as jumping plant lice, which feed on phloem sap [25]. *Ca.* L. asiaticus is transmitted by the citrus psyllid vector *Diaphorina citri* Kuwayama and *Ca.* L. africanus is transmitted by the vector *Trioza erytreae* [4,20]. The primary hosts of *D. citri* are all members of rutaceae family. Major commercial citrus species identified as key hosts include lemon, sour orange, grapefruit, lime, pomelo, mandarin and tangerine. Additionally, *Murraya koenigii* (curry tree) and *M. paniculata* (orange jasmine) are also preferred hosts of this vector [26].

Infected trees exhibit several symptoms which can be observed on the leaves, fruit and canopy of the tree [27]. A higher level of the pathogen is found in the petiole, midrib, peduncle and columella, while a lower level of the pathogen could be observed in seeds, buds and bark [10]. The distinct yellow pattern on a leaf (blotchy mottle), yellowing of leaf veins, premature leaf drop, poor root system, retarded growth are the major symptoms associated with citrus greening disease [28–30]. Fruits from infected trees are often deformed, containing abnormal seeds with increased bitterness and sourness, with reduced sweetness in the juice [10]. Early

HLB symptoms can be observed uniformly across the fibrous roots of infected trees [31]. HLB-infected citrus roots show black to dark brown discoloration due to increased polymerization of lignin and tannins. While this lignification acts as a barrier against pathogen invasion, it also reduces the root's ability to absorb water and nutrients, making it prone to water and temperature stress [31]. Whenever the pathogen infects the phloem tissue, they secrete various effectors and virulence factors and, also triggers plant immune response which is responsible for callose deposition, cell death and protein accumulation, hindering the conductivity of phloem tissue [32].

HLB infects the majority of citrus species, citrus species like grapefruit, sweet oranges, tangelos and mandarins are highly susceptible, while lime, lemon and trifoliate are less susceptible compared to other citrus species [29,33]. There is no confirmed transmission of disease through seed [34]. Trifoliate orange has shown low pathogen titer when grafted to infected rootstock, making it the least susceptible citrus species [35]. HLB is a graft-transmissible disease [36]; side grafts involving twigs are found to be particularly high in pathogen transmission [37]. The grafting method is used to produce citrus saplings in Nepal. Since HLB is highly prevalent, a healthy mother plant is essential to produce citrus saplings. In recent years, PCR has been used for the diagnosis of HLB in mother plants and grafted plants prior to distribution to farmers in Nepal [17].

Appropriate and early diagnosis tools are required to prevent the further spread of disease since bacterial load differs variably depending on several factors [38]. Diagnosis of HLB can be done by several methods like electron microscopy, serology, enzymatic assays, polymerase chain reaction (PCR), quantitative PCR (qPCR) and loop-mediated isothermal amplification coupled with hydroxynaphthol blue (AL-LAMP-HNB) [20,39–42]. PCR technique is widely used for the diagnosis of HLB since it is more sensitive and convenient. Frequently used primers that target 16S rDNA include Las606/LSS [43] and primer OI1/OI2c [44], primer A2/J5 target the nusG-rplK region [45]. Despite the availability of various primers, PCR amplification is not observed even in severely infected trees [46]. It might be due to several factors like a low pathogen in template DNA, the presence of PCR inhibitors and nonspecific primers [47]. Highly conserved regions of the 16S rRNA gene are used for the construction of universal primers and highly variable regions of 16S rRNA are used for the identification of individual species [48]. The reverse primer LSS (5'-ACC CAA CAT CTA GGT AAA AAC C-3') and the forward primer Las606 (5'-GGA GAG GTG AGT GGA ATT CCG A-3') is specific to *Ca.* L. asiaticus. The primer set Las606/LSS was found to be superior to other commonly used primers like A2/J5, OI1/OI2c and MHO353/MHO354. This primer is highly sensitive for conventional PCR, capable of amplifying low template DNA amplifying effectively despite the presence of various contaminants such as ethanol, citric acid, NaCl and sucrose [49].

Phylogenetic analysis is a widely used tool in bioinformatics due to its high reliability and significance [50]. Within a species, phylogenetic analysis aids in unraveling population dynamics, genetic diversity and the evolutionary processes that shape intraspecies variation [51]. The 16S rRNA gene sequence serves as a benchmark for phylogenetic studies. Although being extensively conserved, the existence of interspersed variable regions within the 16S rRNA gene sequence enables the comparison of closely related species [52]. In phylogenetic analysis, the tree is constructed using optimization principles like Maximum Likelihood (ML), Minimum Evolution (ME) and Neighbor Joining (NJ) method [53]. The ML method was employed for molecular identification of the citrus greening pathogen globally [54–56]. The ML method uses statistical models of sequence evolution to infer the tree that best fits the data, although it is more computationally demanding compared to distance-based methods like NJ [57]. In Nepal, HLB was first reported in 1967 in the Pokhara Valley [19]. Since then, the disease has been rapidly observed in different parts of the country, leading to widespread citrus decline [6,17,19,38,58,59]. Many studies on HLB in Nepal have largely relied on surveys and visual assessment [18,38], with only a few employing conventional PCR for confirmation [6,17,60,61]. Therefore, the present research aims to systematically detect *Ca.* L. asiaticus in suspected citrus trees from diverse citrus-growing regions of Nepal using conventional PCR with the Las606/LSS primer set, followed by Sanger sequencing. This approach will provide the molecular data necessary to investigate the origin and phylogenetic relationships of the pathogen, both within Nepal and in a global context.

## Materials and methods

### Sample collection

A total of 1,026 citrus leaf samples were collected over a two-year period (2022–2024) to assess the prevalence of *Ca.* L. asiaticus, the causal agent of Huanglongbing (HLB), in Nepal. All laboratory work was conducted at the Warm Temperate Horticulture Centre (WTHC) in Kirtipur. Samples were methodically collected from major citrus-producing districts across six of Nepal's seven provinces. Madhesh Province was not included in this study because it does not make a significant contribution to citrus production in Nepal [5]. This broad geographical coverage includes- Bhojpur (Koshi Province), Chitwan, Kathmandu, Ramechhap and Sindhuli (Bagmati Province), Gorkha, Lamjung, Myagdi and Syangja (Gandaki Province), Palpa (Lumbini Province), Dailekh, Salyan (Karnali Province) and Dadeldhura (Sudurpashchim Province). This broad sampling strategy was designed to provide a representative overview of the HLB distribution across Nepal. To ensure comprehensive disease surveillance, leaf samples were obtained from both screen houses and open orchards, reflecting different cultivation environments and farming practices. The study included a diverse range of citrus plant materials, incorporating both seedlings and grafted plants. These were collected from various citrus varieties to encompass genetic and phenotypic diversity. Among the different citrus varieties analyzed, mandarin (*C. reticulata*) was the most frequently collected sample, followed by sweet orange (*C. sinensis*), owing to its status as the most widely cultivated citrus crop in Nepal and its economic significance in both commercial and subsistence agriculture across the country. The study incorporated other citrus varieties, including acid lime, pomelo, unshu mandarin, navel orange and kumquat. It encompassed local citrus varieties which are native to Nepal, such as khoku mandarin and banskharka mandarin. Furthermore, newly introduced varieties in the country, such as Valencia Late Orange, Avana Apireno mandarin, Imperial mandarin, Daisy tangerine and Washington Navel orange were also included in the study.

### Sample management

The sampling procedure followed the guidelines outlined in HLB-SOP-1, established by the Ministry of Agriculture and Livestock Development, Government of Nepal [62], ensuring a standardized and systematic approach to sample collection. Citrus trees were visually inspected for characteristic symptoms of HLB, such as blotchy mottle patterns and leaf chlorosis. From each tree, fully expanded leaves were collected from multiple directions below the canopy to account for potential variability in bacterial distribution. Sampling was conducted in the August–November period, with a preference for symptomatic leaves when available.

Immediately after collection, samples were placed in an ice box to maintain their integrity during transport to the laboratory. When feasible, sampling was conducted during the early morning or evening to minimize degradation. All samples were transported to the laboratory within two days of collection. Upon arrival at the laboratory, samples were stored at 4°C for a maximum of one day before processing. For processing, leaf samples were thoroughly cleaned to remove surface contaminants. The mid-vein sections were then excised, cut into small pieces frozen in liquid nitrogen and subsequently stored in a deep freezer at −35°C until DNA extraction.

### DNA isolation and visualization

DNA isolation was performed using the CTAB method [63] with slight modifications to extract high-quality genomic DNA from citrus leaf samples. Approximately 0.2 g of the mid-vein section from each leaf sample was carefully ground into a fine powder using a mortar and pestle in the presence of liquid nitrogen. This step was essential to facilitate cell disruption and ensure the release of cellular content. Once the tissue was finely ground, it was mixed with 1 ml of CTAB buffer, which was composed of 2% CTAB, 0.5 M EDTA, 5 M NaCl, 1 M Tris-HCl and 0.2% PVP. The resulting mixture was formed into a fine paste, which was then transferred into a 1.5 ml microcentrifuge tube for further processing. The sample was incubated at 65°C in a water bath for 45 minutes to enable the lysis of cells and the release of DNA. During this incubation

period, the sample was gently mixed every 10 minutes to ensure uniform cell disruption and complete DNA release. After the incubation step, the sample was centrifuged at 12,000 rpm for 8 minutes to pellet any cellular debris. The supernatant, containing the DNA, was carefully transferred (approximately 600 μl) into a clean, sterile 2 ml microcentrifuge tube. To remove proteins and other contaminants, an equal volume of Chloroform: Isoamyl alcohol (24:1) was added to the supernatant. The solution was then mixed gently on an orbital shaker for 15 minutes to facilitate phase separation. After mixing, the sample was centrifuged at 13,000 rpm for 5 minutes to separate the phases. The upper aqueous phase, containing the DNA, was carefully transferred (approximately 400 μl) into a sterile 1.5 ml microcentrifuge tube. To precipitate the DNA, 40 μl of 3M sodium acetate was added, followed by 500 μl of ice-cold absolute ethanol. The mixture was then centrifuged at 13,000 rpm for 2 minutes, which led to the formation of a DNA pellet. The supernatant was discarded and the DNA pellet was washed with 500 μl of ice-cold 70% ethanol. The sample was then centrifuged again at 13,000 rpm for 1 minute to remove any residual salts. This washing step was repeated twice to ensure the thorough removal of contaminants, resulting in a high-quality DNA sample. After the final wash, the ethanol was carefully pipetted out and the DNA pellet was dried at 37°C for 30 minutes in an incubator. Once dried, the DNA was suspended in 100 μl of 1X TE buffer and stored at −35°C for further use. To confirm the quality of the isolated DNA, it was visualized on a 0.8% agarose gel and the DNA bands were observed using a gel documentation system to verify the integrity and purity of the extracted DNA. This step ensured that the DNA was suitable for downstream applications, including PCR and sequencing.

### HLB diagnosis using conventional Polymerase Chain Reaction (PCR)

The extracted DNA was used for the detection of *Ca.* L. asiaticus via conventional PCR. Amplification targeted a 500 bp region of the pathogen's genome using the specific primer pair Las606 and LSS [49]. Table 1 shows the sequence of the primers used in this study. PCR was performed in a Techne TC-312 thermal cycler. Each PCR reaction mixture (15 μl) contained 2 μl of template DNA, 0.3 μl of forward primer, 0.3 μl of reverse primer, 6.5 μl of GoTaq Green Master Mix (Promega Corporation) and 5.9 μl of nuclease-free water. The GoTaq Green master mix is a premixed solution that includes Taq DNA polymerase, dNTPs, $MgCl_2$ and optimized reaction buffers, making it ideal for efficient DNA amplification. Additionally, the mix contains two dyes (blue and yellow) that allow for real-time visual monitoring of the PCR progress during electrophoresis. After amplification, the PCR products were visualized on a 2% agarose gel. A 100 bp ladder marker (Solis BioDyne) was run alongside the samples to determine the size of the PCR products. A distinct band at 500 bp was scored as a positive detection of *Ca.* L. asiaticus. To ensure diagnostic reliability, each PCR run included a previously confirmed positive control and a negative (nuclease-free water) control. All reactions were performed in triplicate to verify the consistency and reproducibility of the results.

### Sequencing and phylogenetic analysis

**Sequencing of positive samples and Initial BLAST Analysis.** PCR products from five positive samples, representing five major citrus-growing provinces (Bagmati, Gandaki, Lumbini, Karnali and Sudurpashchim), were sent to the Center for Molecular Dynamics Nepal (CMDN) for Sanger sequencing. Despite Koshi province being one of the major citrus producers, its sample was not included because of low-quality DNA. The details of citrus samples selected for sequencing across various provinces of Nepal are presented in Table 2. Both forward and reverse primers were used for sequencing to obtain sequence information for phylogenetic analysis. The ten raw sequences generated from these five

**Table 1. Primers used for PCR amplification.**

| S.N. | Primer name | Sequence |
|------|-------------|----------|
| 1. | Forward primer Las606 | 5'-GGA GAG GTG AGT GGA ATT CCG A-3' |
| 2. | Reverse primer LSS | 5'-ACC CAA CAT CTA GGT AAA AAC C-3' |

**Table 2. Details of citrus samples selected for sequencing across different provinces of Nepal.**

| S. N. | Province name | Sample selected (district) | Sample ID | Symptom presence | DNA quality |
|---|---|---|---|---|---|
| 1. | Koshi province | None | none | Asymptomatic | Poor |
| 2. | Bagmati province | Kathmandu | WTHC_05 | Symptomatic | Good |
| 3. | Gandaki province | Myagdi | WTHC_02 | Symptomatic | Good |
| 4. | Lumbini province | Palpa | WTHC_01 | Symptomatic | Good |
| 5. | Karnali province | Dailekh | WTHC_03 | Symptomatic | Good |
| 6. | Sudurpashchim province | Dadeldhura | WTHC_04 | Symptomatic | Good |

a One representative sample per province was selected for sequencing. Koshi Province was excluded from sequencing due to the poor DNA quality and a low-intensity amplification signal in the PCR assay, rendering the sample unsuitable for downstream molecular analysis.

samples were submitted to the NCBI database. GenBank accession numbers PP916596–PP916605 were assigned to the raw sequences for further reference and analysis.

**Consensus sequence generation and phylogenetic analysis.** The quality of the raw sequences was analyzed by examining the chromatogram peaks using SnapGene software. A single consensus sequence was generated by aligning the forward and reverse sequences through AliView software [64]. For this purpose, forward and reverse gene sequences of *Ca.* L. asiaticus obtained from samples collected in Kathmandu, Dadeldhura, Dailekh, Myagdi and Palpa, were merged to generate a single consensus sequence. Using the AliView software. Specifically, the following GenBank accession pairs were merged: PP916596 with PP916601, PP916597 with PP916602, PP916598 with PP916603, PP916599 with PP916604 and PP916600 with PP916605 to generate single consensus sequences for each sample as shown in Table 3, which were then used for subsequent phylogenetic analysis. A phylogenetic tree was constructed using the maximum likelihood method with 500 bootstrap replications and this analysis was performed using MEGA 11 software [65].

## Results

### Sample information

Leaf samples were collected from citrus orchards located in various geographical regions of Nepal (Fig 1). During the field collection, several trees displayed characteristic visual symptoms of citrus greening like yellowing and blotchy mottle pattern on the leaves (Fig 2). In addition, infected trees showed poor growth with small fruit size. A few asymptomatic trees with green leaves were also tested to be HLB positive.

**Table 3. GenBank accession numbers and consensus sequences of *Candidatus* Liberibacter asiaticus obtained in this study.**

| S.N. | GenBank Accession Number | Sequence Name | Consensus sequence |
|---|---|---|---|
| 1. | PP916601 | WTHC_05_Clas_Kathmandu_Nepal_FWD | PP916601.1 merged PP916596.1 CaLas Kathmandu Nepal |
| 2. | PP916596 | WTHC_05_Clas_Kathmandu_Nepal_RVS | |
| 3. | PP916602 | WTHC_04_Clas_Dadeldhura_Nepal_FWD | PP916602.1 merged PP916597.1 CaLas Dadeldhura Nepal |
| 4. | PP916597 | WTHC_04_Clas_Dadeldhura_Nepal_RVS | |
| 5. | PP916603 | WTHC_03_Clas_Dailekh_Nepal_FWD | PP916603.1 merged PP916598.1 CaLas Dailekh Nepal |
| 6. | PP916598 | WTHC_03_Clas_Dailekh_Nepal_RVS | |
| 7. | PP916604 | WTHC_02_Clas_Myagdi_Nepal_FWD | PP916604.1 merged PP916599.1 CaLas Myagdi Nepal |
| 8. | PP916599 | WTHC_02_Clas_Myagdi_Nepal_RVS | |
| 9. | PP916605 | WTHC_01_Clas_Palpa_Nepal_FWD | PP916605.1 merged PP916600.1 CaLas Palpa Nepal |
| 10. | PP916600 | WTHC_01_Clas_Palpa_Nepal_RVS | |

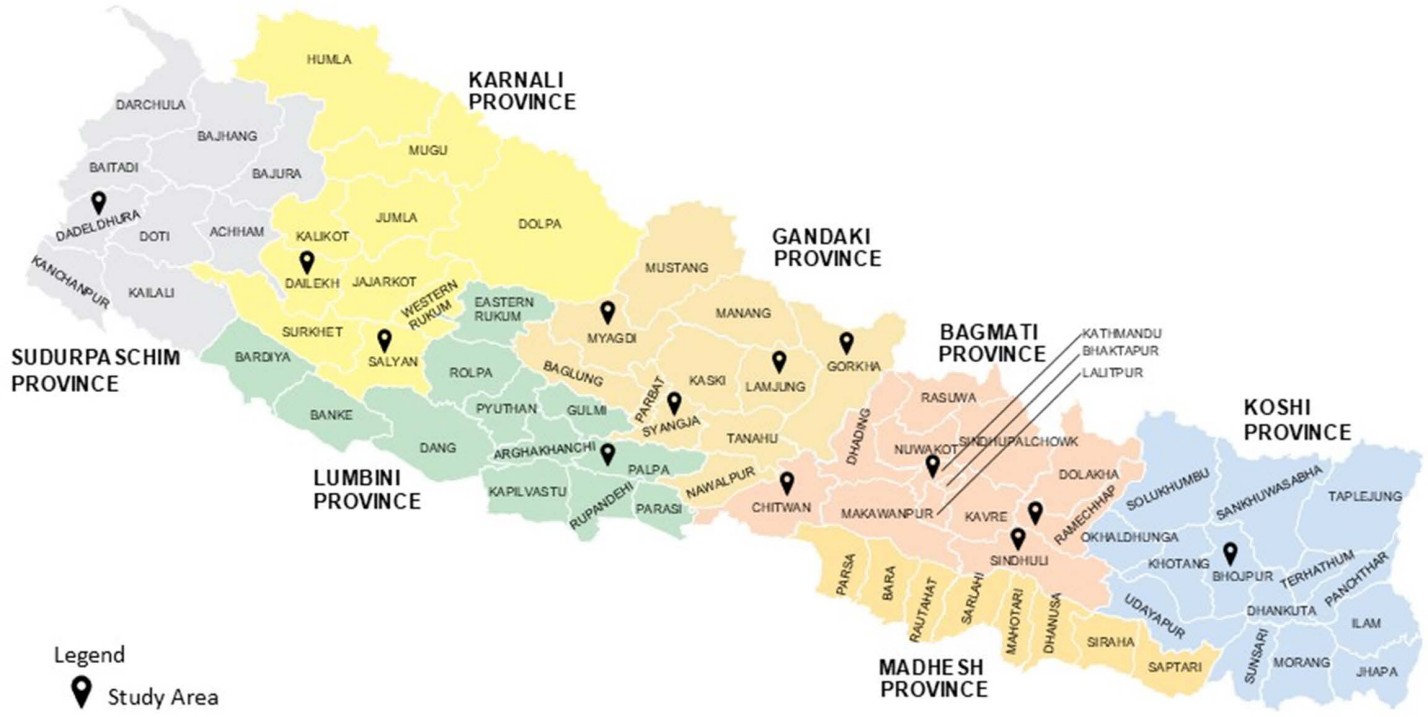

**Fig 1. Map of Nepal indicating citrus sample collection sites across different provinces for molecular detection of HLB. Sampling was carried out in selected districts from Provinces 1, 3, 4, 5, 6 and 7.**

## DNA extraction and PCR amplification for diagnosis of disease

DNA extraction of all 1026 samples was performed using the CTAB method with slight modifications. After DNA extraction, the quality of the DNA was assessed by visualizing it through gel electrophoresis (Fig 3). Conventional PCR amplification was carried out using the primer set Las606/LSS, targeting the 16S rRNA gene of *Candidatus* Liberibacter asiaticus. Among the 1026 samples collected from various geographical regions of Nepal, 255 samples tested positive for HLB, validated by the presence of a 500 bp amplicon (Fig 4). The result of conventional PCR indicates the prevalence of HLB in multiple areas of Nepal, with significant regional variation. Among the samples tested, the maximum number of positive samples was detected from Ramechhap (74 samples), followed by Gorkha district (43 samples) and Sindhuli (39 samples), sample from Kathmandu and Syangja districts showed low HLB infection (Fig 5). The blue bars represent HLB-positive samples, indicating the presence of the citrus greening pathogen, whereas the orange bars represent HLB-negative samples.

## Sequencing and phylogenetic analysis

The phylogenetic tree was constructed to analyze the genetic relationships among the *Ca.* L. asiaticus strains identified in the study, based on the sequences obtained through PCR amplification. The Maximum Likelihood (ML) method, a character-based algorithm, was used to construct the tree. The ML method infers the evolutionary relationship by finding the tree under a specific model of evolution. All DNA Sequences from Nepal showed strong evolutionary relatedness within the *Ca.* L. asiaticus cluster, displaying over 99% genetic similarity with the distant sequences from different parts of the world, indicating a possible common origin for these strains. The Basic Local Alignment

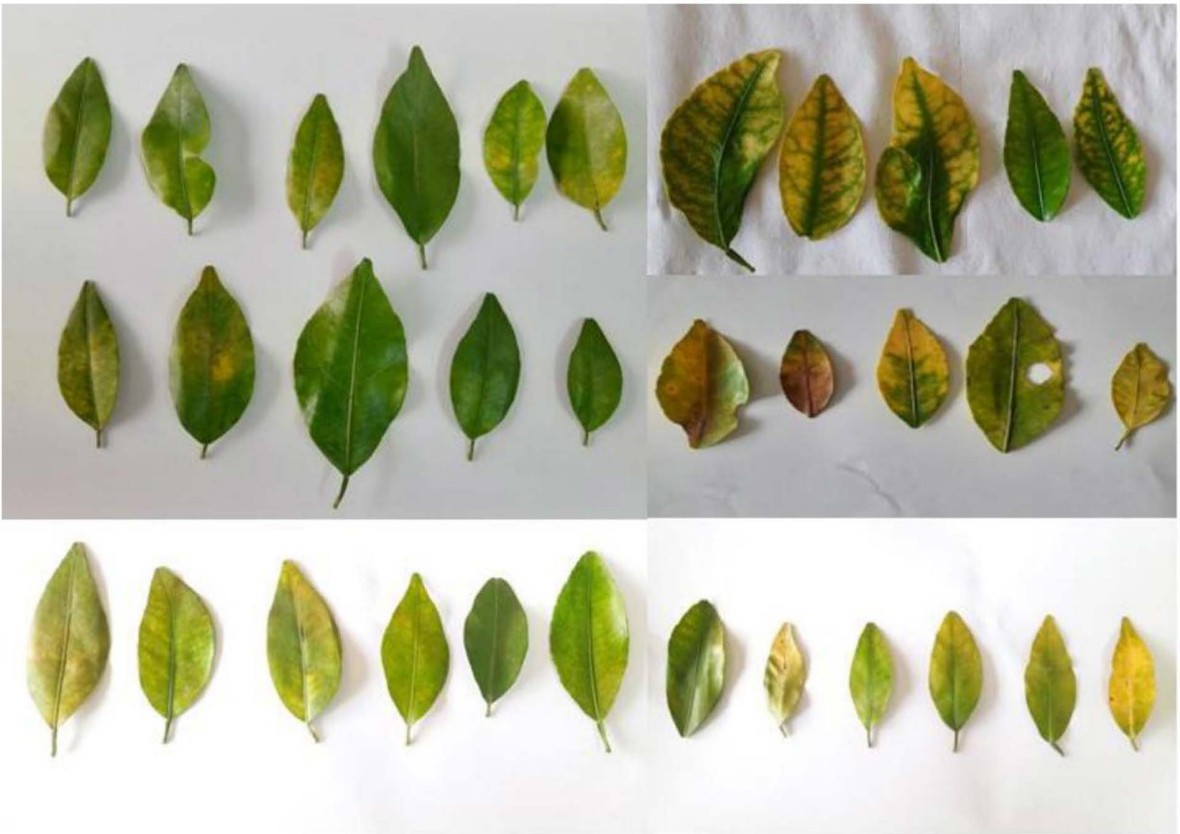

**Fig 2. Positive leaf samples exhibiting visual symptoms of citrus greening disease like blotchy mottle and yellowing of leaves.**

Search Tool (BLAST) homology search analysis was executed to assess sequence similarity with *Ca*. L. asiaticus from various geographical regions. Consensus sequences amplified using Las606/LSS showed greater than 99% homology with *Ca*. L. asiaticus 16S rRNA gene sequences: JQ867410.1 (Yucatan, Mexico), OM522080.1 (Punjab, India), OP106890.1 (Saudi Arabia), CP159585.1 (Guangzhou, China), MH473397.1 (Meerut, India), MH473394.1 (Meerut, India), PP333207.1 (Tamilnadu, India), MK142763.1 (USA), MN575695.1 (Trinidad, West Indies). Additionally, 16S rRNA gene sequences of *Ca*. L. africanus (KU561669.1 from South Africa, LN795908.1 from Madagascar) and *Ca*. L. americanus (OQ725634.1 and OQ725635.1 from India, AY742824.1 from São Paulo, Brazil) were used to construct a phylogenetic tree. The 16S rRNA gene sequence of *Acholeplasma palmae* (L33734.1) served as an outgroup.

The phylogenetic tree revealed distinct clades of *Ca*. L. asiaticus, *Ca*. L. africanus and *Ca*. L. americanus sequences from various geographic regions and significant branches were supported by high bootstrap values, indicating strong confidence in those groupings. The 16S rRNA gene sequences from different parts of Nepal were clustered in the same clade with *Ca*. L. asiaticus sequences obtained from different parts of the world. Within this clade, a subcluster of sequences originating from Tamilnadu, India (PP333207.1), Trinidad, West Indies (MN473394.1) and Guangzhou, China (CP159585.1) were observed with PP333207.1 and CP159585.1 exhibiting a close relationship to one another when compared to MN473394.1. Additionally, 16S rRNA gene sequences of *Ca*. L. africanus (KU561669.1 from South Africa and LN795908.1 from Madagascar) and *Ca*. L. americanus (OQ725634.1, OQ725635.1 from India and AY742824.1 from Sao Paulo, Brazil) were grouped into separate clades. The *Ca*. L.

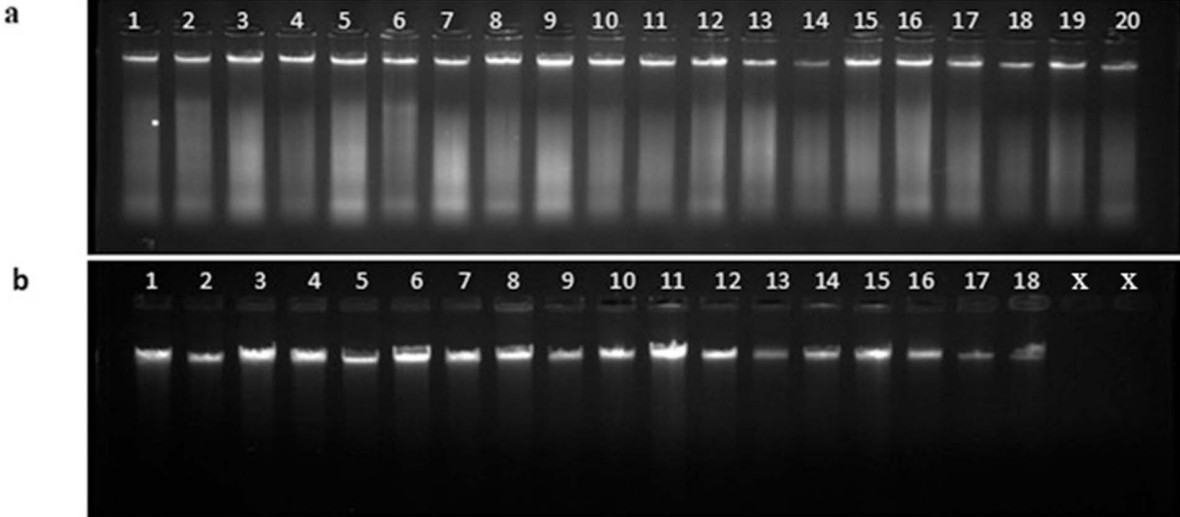

**Fig 3. Agarose gel electrophoresis using 0.8% agarose for visualization of DNA extracted from the mid-vein of citrus leaves collected from various regions of Nepal. (a)** DNA extracted from citrus leaf mid-veins. **(b)** DNA extracted from citrus leaf mid-veins from additional samples.

africanus sequences showed closer evolutionary relatedness to Ca. L. asiaticus compared to *Ca*. L. americanus, which displayed greater divergence from *Ca*. L. asiaticus (Fig 6).

## Discussion

Leaf samples were thoroughly inspected for visual signs and symptoms of HLB before DNA extraction. While some positive samples exhibited the characteristic blotchy mottle pattern and vein yellowing, most leaf samples did not show the classic symptoms of HLB. Instead, these samples displayed symptoms like those caused by nutrient deficiencies. This observation underscores the unreliability of using visual symptoms alone for the diagnosis of HLB, as HLB-positive leaves can exhibit symptoms that overlap with those of other diseases, such as citrus tristeza virus and can also resemble nutrient deficiencies [22]. It is highly unreliable to perform disease diagnosis based only on visual symptoms since the disease might remain asymptomatic due to uneven distribution of bacteria [66,67]. PCR-based molecular detection offers a more reliable and accurate method for diagnosing citrus greening disease [61,68].

A total of 1,026 plant samples were analyzed to assess pathogen prevalence across various districts in Nepal. Among these, 255 samples (24.85%) were found to be positive, while 771 samples (75.15%) were found to be negative, indicating a significant variation in disease incidence across different geographical regions. A bar chart was constructed for the number of citrus leaf samples tested for citrus greening disease across different districts of Nepal (Fig 5). The highest positive cases were observed in Ramechhap district (74/197, 37.56%), followed by Gorkha district (43/87, 49.42%) and Sindhuli district (39/209, 18.66%), suggesting these areas as potential hot spots for pathogen transmission. Similarly, Myagdi district (9/15, 60%) and Lamjung district (6/9, 66.67%) also showed high positivity rates; however, the small sample sizes in these districts necessitated further testing to confirm the actual disease prevalence. In contrast, Kathmandu district (1/37, 2.70%), Syangja district (1/46, 2.17%) and Dadeldhura district (6/187, 3.21%) exhibited relatively low positive cases, suggesting either lower pathogen prevalence or successful disease control measures in these areas. Bhojpur district (10/20, 50%) and Chitwan district (9/10, 90%) had high proportions of positive cases, but the small number of samples limited the accuracy of disease prevalence estimation. To gain a more comprehensive understanding of pathogen distribution, it is essential to increase sample size in districts with limited samples,

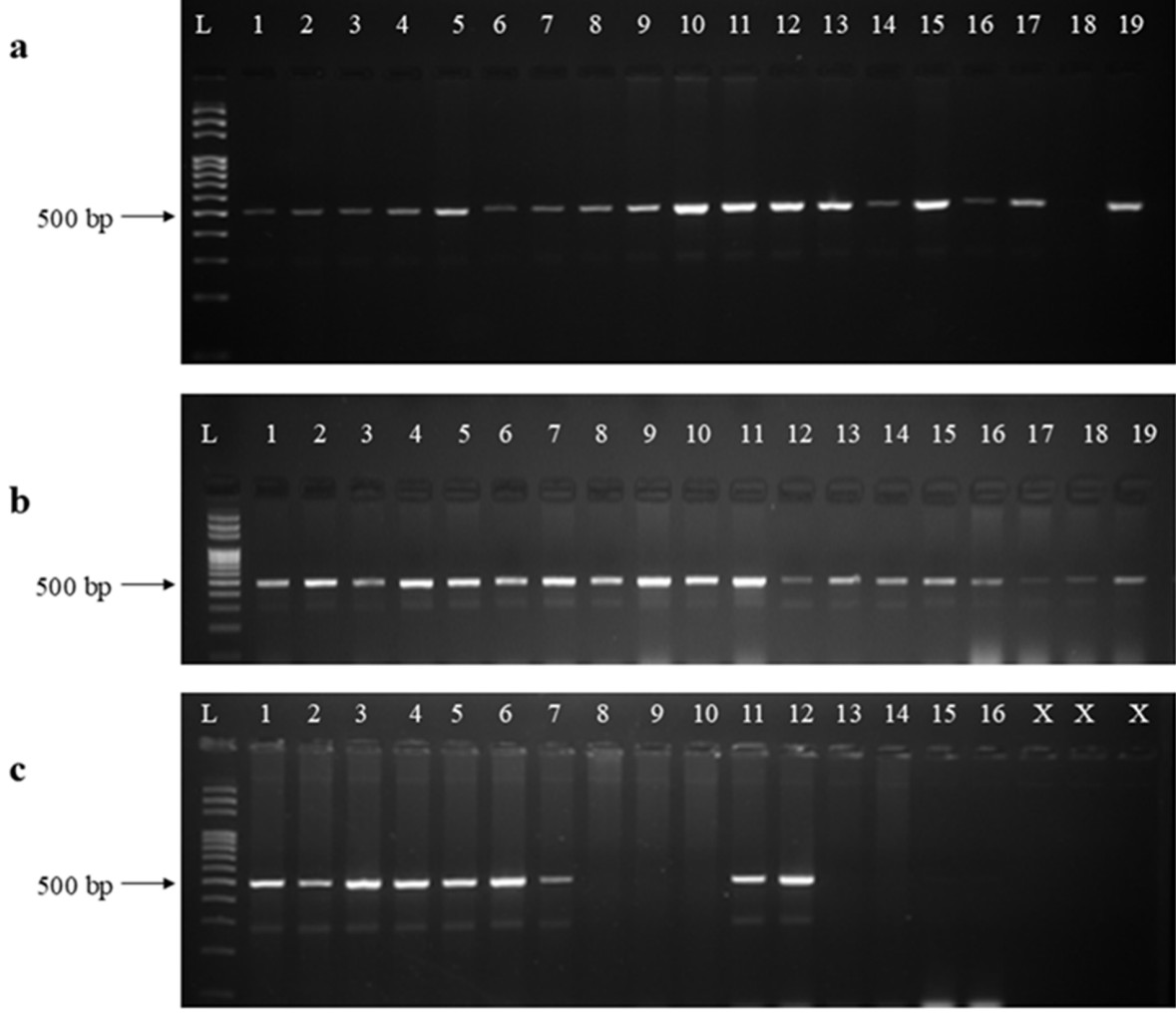

**Fig 4. Agarose gel electrophoresis using 2% agarose for visualization of PCR products amplified with the primer set Las606/LSS. Positive samples show a 500 bp amplicon, while negative samples show no amplification.** (L) ladder **(a)** PCR amplification of HLB-positive samples (well 1–17), negative control (well 18) and positive control (well 19). **(b)** PCR amplification of additional HLB-positive samples (well 1–19). **(c)** PCR amplification of both HLB-positive and negative samples.

such as Chitwan district, Bhojpur district, Lamjung district and Myagdi district, to ensure statistically significant disease incidence data. The samples for this study were sourced from major citrus-producing regions of Nepal, highlighting a growing concern about the potential decline in citrus orchards due to the spread of HLB. The spread of disease across Nepal may be attributed to grafting with uncertified rootstocks and scions, as well as the improper management of citrus orchards. In an earlier study, Oliya [6] detected *Ca.* L. asiaticus in various citrus species, except pomelo (*C. grandis*), from Lamjung and Gorkha districts; however, no infection was detected in samples from Dolakha district. A subsequent study conducted in 2024 reanalyzed *Ca.* L. asiaticus in mandarin orange samples from the same orchards in Gorkha, Lamjung and Dolakha and confirmed the presence of the pathogen at all three locations [17]. In another investigation, seventy-nine citrus samples were examined using PCR with the OI1/OI2C and OAI1/OI2C primer sets and seventeen samples were confirmed positive [60]. Similarly, the presence of the pathogen was confirmed in different

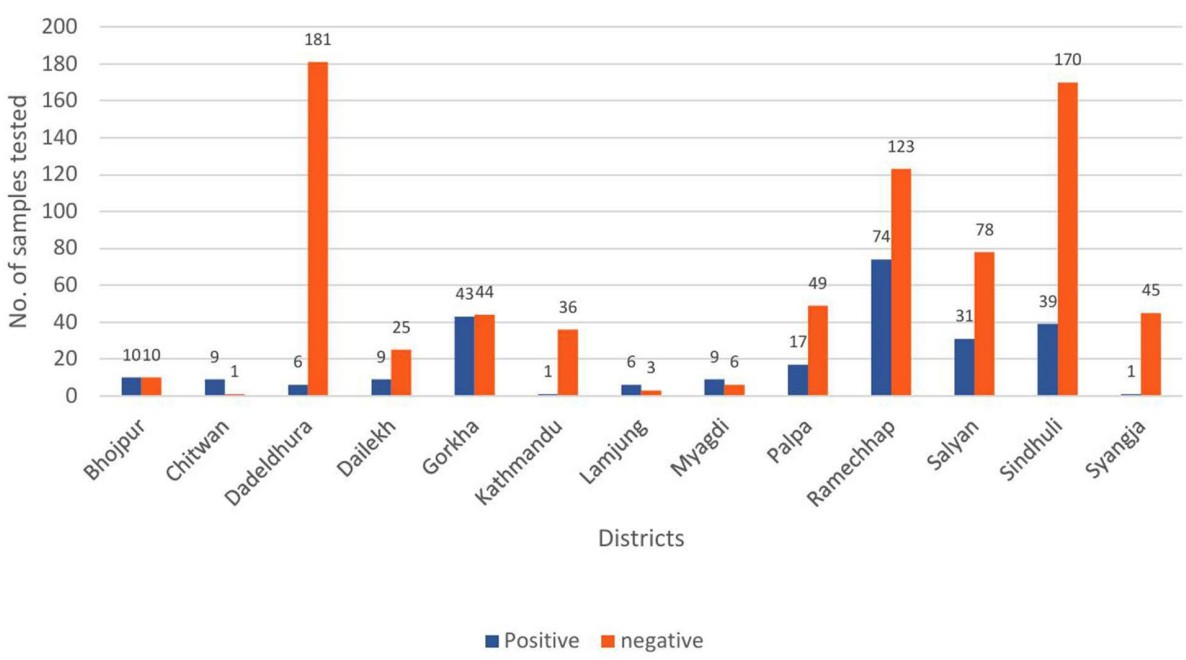

**Fig 5. Bar chart showing the number of citrus leaf samples tested for citrus greening disease using PCR across different districts of Nepal.** The blue bars represent HLB-positive samples indicating the presence of the citrus greening pathogen, while the orange bars represent HLB-negative samples.

regions using conventional PCR and loop-mediated isothermal amplification (LAMP) methods [69]. These findings high-light the progressive spread of *Ca.* L. asiaticus across different citrus growing regions of the country and underscores the urgent need for reliable diagnostic tools. Given that HLB poses a serious threat to citrus production, the choice of detection method is critical for effective management.

In Nepal, conventional PCR remains the primary diagnostic tool for confirming *Ca.* L. asiaticus, mainly due to its cost-effectiveness, high sensitivity and specificity. The standard protocol involves extracting genomic DNA from leaf tissue (often using a kit), followed by amplification with specific primers and gel electrophoresis to confirm the presence of patho-gen DNA. However, conventional PCR has notable limitations: its effectiveness can be limited by the uneven distribution of the pathogen within infected plants, as the bacteria may not be uniformly present across all tissues [70]. This may result in false-negative outcomes if the sampled tissue does not contain the bacterium. To overcome this limitation, Nepal should consider developing a real-time PCR–based detection system. This advanced technique not only enhances detection sensitivity but also facilitates the identification of the pathogen in both leaf and root tissues [40]. By targeting multiple plant parts, real-time PCR can detect infections even when the pathogen is unevenly distributed, thereby enabling earlier diagnosis and more accurate surveillance of HLB in citrus orchards [43,49]. Recent developments include new qPCR protocols validated according to international standards, such as those from the European Plant Protection Organization (EPPO), which can detect all pathogenic *Ca.* Liberibacter species in a single assay with high specificity [39]. A new duplex PCR-lateral flow immunoassay (d-PCR-LFIA) has been developed where a duplex PCR detects both *Candidatus* Liberib-acter asiaticus and Citrus tristeza virus in a single sample while LFIA enables on-site diagnosis without gel electrophoresis making it a practical alternative to conventional PCR [11].

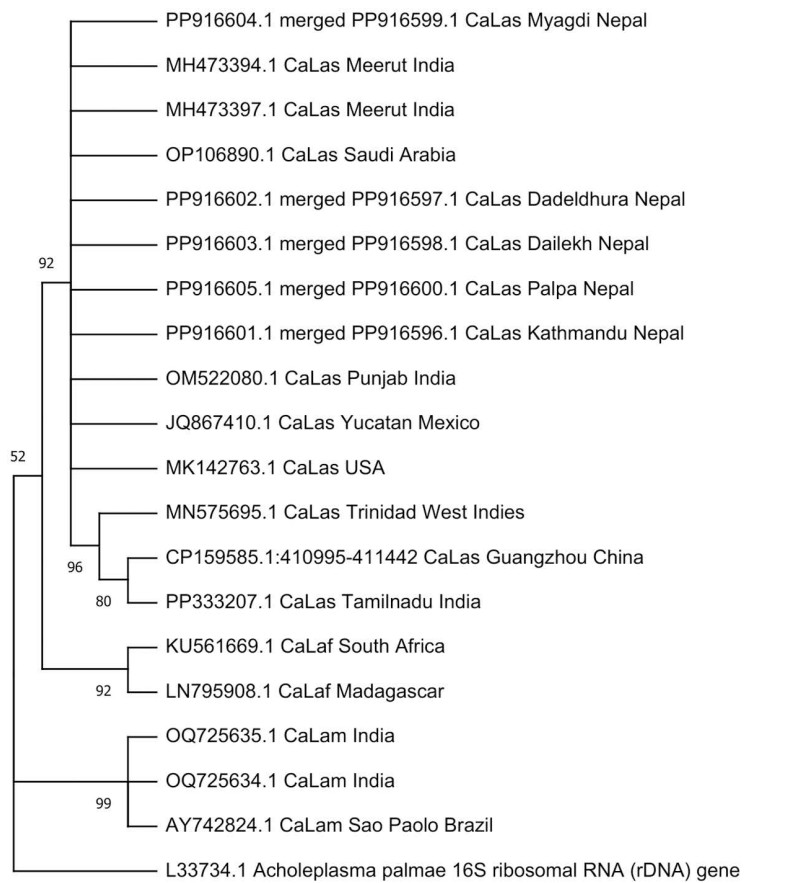

**Fig 6. Phylogenetic analysis of 16S rDNA sequences amplified with Las606/LSS primers from five provinces of Nepal and other reported** *Candidatus* **Liberibacter species using the Maximum Likelihood algorithm under the General Time Reversible model with gamma-distributed rate variation and a proportion of invariant sites (GTR+G+I) algorithm.** The phylogeny was tested using 500 bootstrap replicates. L33724.1 *Acholeplasma palmame* 16S ribosomal RNA gene was used as an outgroup.

In this study, the diagnosis of all samples was performed using the primer set Las606/LSS, which amplifies a 500 bp segment of the 16S rRNA gene of *Ca.* L. asiaticus. This primer set is particularly sensitive compared to the previously documented primers, including A2/J5 and OI1/OI2c [49,54]. In a study of a symptomatic Kinnow mandarin orchard in India, Shweta et al. [54] evaluated two primer sets, Las606/LSS and OI1/OI2c, for HLB detection. Using the primer pair Las606/LSS, they obtained positive PCR amplification in 40 out of 57 samples. In contrast, the OI1/OI2c primers were positive in only two samples. This result also demonstrates the superior sensitivity and specificity of the Las606/LSS primer set for detecting *Ca.* L. asiaticus [54]. One significant advantage of the Las606/LSS primer set is its ability to remain effective even in the presence of contaminants like ethanol and starch, which can inhibit PCR amplification. This makes it a more robust option for field samples that may contain such impurities. Moreover, this primer set is highly sensitive and capable of detecting as low as 1 ng of DNA, ensuring accurate detection even in samples with low concentrations of the pathogen [43]. These characteristics make the Las606/LSS primer set an excellent choice for the molecular detection of *Ca.* L. asiaticus, offering both reliability and sensitivity in diagnostic applications.

In the present study, HLB-positive samples consistently produced approximately 500 bp PCR amplicon, which was visualized through agarose gel electrophoresis, confirming the presence of *Ca.* L. asiaticus. In contrast, no amplification was observed in HLB-negative samples, indicating the absence of the target pathogen. Each PCR assay was carefully

designed to include a positive control, which was obtained from a previously confirmed DNA sample, ensuring that the reaction conditions were optimal and the primers were functioning correctly. Additionally, a negative control was included in the reaction setup to detect any potential contamination in the PCR master mix or reagents. To enhance the reliability of the results, each PCR reaction was performed in triplicate. Moreover, each gel electrophoresis included a DNA ladder to ensure accurate determination of PCR product sizes.

The primary objective of this study was to achieve the molecular confirmation and initial genetic characterization of *Ca.* L. asiaticus, a country where no prior pathogen sequence data existed. Therefore, a total of five candidate samples representing different geographical regions of Nepal were selected and submitted for Sanger sequencing. Sanger sequencing is widely used in the sequencing field as it offers several prominent advantages: cost-efficiency for sequencing single genes and 99.99% accuracy [71]. DNA sequencing was performed using both forward (Las606) and reverse primer (LSS) sequences. Forward and reverse primers are necessary for Sanger sequencing to ensure accurate and complete sequencing of the target DNA fragment. Moreover, the bidirectional sequencing helps to identify and resolve any potential ambiguities or mutations present in the target sequence. Thus, by comparing the sequences obtained from both directions, high-quality sequencing data can be generated, which facilitates accurate interpretation of the results [72]. The evolutionary relatedness of the sequences from different geographical regions can be assessed by analyzing their position within the phylogenetic tree.

In the present study, the BLAST analysis of the 16S rDNA gene confirmed the identity of the Nepalese isolates as *Ca.* L. asiaticus with >99% sequence identity. Notably, sequences from different regions of Nepal clustered closely together, indicating high genetic homogeneity and limited diversity within the national *Ca.* L. asiaticus population. This limited diversity could be attributed to the sampling of a relatively restricted number of isolates from similar agro-ecological conditions, or it may suggest a recent introduction and clonal expansion of a single predominant strain within Nepal [73]. A large-scale study examining 500 *Ca.* L. asiaticus samples from across China reported a wide distribution and high diversity of associated prophages [74]. Their work highlights how climate change may influence future pathogen distribution. Together, such insights are critical for shaping effective huanglongbing management strategies, as understanding both local population dynamics and large-scale ecological drivers can facilitate the development of measures to control this devastating disease.

Phylogenetically, the Nepalese sequences showed the closest evolutionary relationship to isolates from neighboring India (OM522080.1 from Punjab and MH473394.1 from Meerut), supporting the hypothesis of pathogen movement across the shared border [15]. This pattern is biologically reasonable, considering the geographical proximity between Nepal and India, frequent cross-border movement of plant materials and shared agro-ecological conditions that favor pathogen establishment and persistence [14]. The close association suggests that Nepalese CaLas populations may have originated from or share ancestry with Indian populations, supporting the idea of regional spread rather than independent evolution. Additionally, Nepalese isolates are positioned within a broader clade that includes sequences from China, Saudi Arabia, the USA, Mexico and the Caribbean. The absence of clear geographic structuring within this clade indicates that global CaLas populations share a high degree of genetic similarity.

Our finding that the Nepalese and Indian (Meerut) isolates cluster closely aligns with the broader phylogeographic pattern of *Ca*. L. asiaticus. Previous studies have shown that the Meerut isolate itself groups with strains from China, Iran, Canada and Italy [75], underscoring the global, yet regionally structured, distribution of this strain. This pattern was first established by Ding et al. [76], who molecularly classified Chinese HLB isolates as *Ca*. L. asiaticus and demonstrated its prevalence across a wide geographical range, including India, Japan, Thailand and North America, while distinguishing it from African *Ca*. L. africanus.

The sequences from Nepal, while showing strong intra-country relatedness, clustered definitively within the global *Ca.* L. asiaticus clade, sharing >99% genetic identity with isolates from geographically distant regions. This high degree of conservation suggests a common origin or recent shared ancestry for these strains, consistent with patterns observed in

India and China [75,77]. It is plausible that *Ca*. L. asiaticus strains with a common evolutionary lineage were introduced into Nepal through global trade networks or natural vector dispersal mechanisms. Overall, the close evolutionary relations between *Ca*. L. asiaticus sequences from Nepal and those from around the world highlight the interconnectedness of citrus cultivation globally and the ease with which this pathogen transcends borders. A clear understanding of these transmission dynamics is therefore essential for formulating coordinated, regional management strategies to mitigate the spread and impact of HLB in Nepal and across susceptible citrus-producing regions worldwide. To gain better insights into genetic diversity and intra-species variation across Nepal, future studies should employ multi-locus or whole-genome sequencing approaches with broader geographic sampling.

## Conclusions

This study represents the first extensive diagnosis and initial phylogenetic assessment of *Candidatus* Liberibacter asiaticus in Nepal, a context in which no prior pathogen sequence data were available. The findings reveal widespread prevalence of citrus greening disease across Nepal, which can be attributed to ineffective management practices, the use of uncertified planting materials, uniform mid-hill climatic conditions and inadequate vector management. The confirmation of the pathogen in Nepal provides a scientific foundation for establishing routine surveillance systems and integrating PCR screening into national monitoring programs. Furthermore, phylogenetic analysis of 16S rDNA sequences revealed both regional clustering (Nepal–India linkage) and global genetic homogeneity. These results emphasize the necessity of advanced molecular diagnostics and genetic analyses, along with the urgent implementation of integrated management policies including the implementation of nursery certification programs for disease-free planting materials, and adoption of targeted management strategies, including vector control and removal of infected plants. The study further supports policy-level actions such as investment in diagnostic infrastructure, development of management guidelines, and enhanced quarantine measures. Additionally, the generated sequence data also contribute to global pathogen databases, enhancing understanding of the distribution and genetic relationships of *Candidatus* Liberibacter asiaticus.

## Supporting information

**S1 File. Sequence data that support the findings of this study have been deposited in the National Center for Biotechnology Information (NCBI) with primary accession code PP916596 to PP916605.**
(PDF)

**S2 File. Consensus sequence used in the construction of phylogenetic tree generated using SnapGene and AliView software.**
(PDF)

**S3 File. Raw images. Original images for gel.**
(PDF)

## Acknowledgments

We thank the field staff and farmers for their assistance in sample collection and Ms. Sunita Khadka for laboratory support. We acknowledge the Warm Temperate Horticulture Center, National Center of Fruit Development, Ministry of Agriculture and Livestock Development, Government of Nepal, for their support to conduct this research work.

## Author contributions

**Conceptualization:** Bal Kumari Oliya.

**Data curation:** Richa Giri, Bal Kumari Oliya.

**Formal analysis:** Bal Kumari Oliya.

**Investigation:** Richa Giri, Bal Kumari Oliya.

**Methodology:** Bal Kumari Oliya, Siddartha Gautam.

**Software:** Bal Kumari Oliya, Siddartha Gautam.

**Supervision:** Bal Kumari Oliya, Krishna Das Manandhar.

**Validation:** Bal Kumari Oliya.

**Visualization:** Richa Giri.

**Writing – original draft:** Richa Giri, Bal Kumari Oliya.

**Writing – review & editing:** Bal Kumari Oliya, Krishna Das Manandhar.

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
