## [Decision Letter · Decision Letter 0]

12 Nov 2025

PONE-D-25-50808PCR-Based Detection and Phylogenetic Analysis of Candidatus Liberibacter asiaticus in Citrus Orchards Across NepalPLOS ONE

Dear Dr. Oliya,

Thank you for submitting your manuscript to PLOS ONE. After careful consideration, we feel that it has merit but does not fully meet PLOS ONE’s publication criteria as it currently stands. Therefore, we invite you to submit a revised version of the manuscript that addresses the points raised during the review process.

Please submit your revised manuscript by Dec 27 2025 11:59PM. If you will need more time than this to complete your revisions, please reply to this message or contact the journal office at plosone@plos.org. Please include the following items when submitting your revised manuscript:

We look forward to receiving your revised manuscript.

Kind regards,

Nishant Kumar, Ph.D

Academic Editor

PLOS ONE

Journal Requirements:

[The authors have declared that no competing interests exist.].

We note that one or more of the authors are employed by a commercial company: Himalayan Plants Solution.

4. Please amend the manuscript submission data (via Edit Submission) to include author Siddartha Gautam.

5. Please amend your authorship list in your manuscript file to include author Siddartha Gautam Gautam.

6. We note that Figure 1 in your submission contains map images which may be copyrighted. All PLOS content is published under the Creative Commons Attribution License (CC BY 4.0), which means that the manuscript, images, and Supporting Information files will be freely available online, and any third party is permitted to access, download, copy, distribute, and use these materials in any way, even commercially, with proper attribution. For these reasons, we cannot publish previously copyrighted maps or satellite images created using proprietary data, such as Google software (Google Maps, Street View, and Earth). For more information, see our copyright guidelines: http://journals.plos.org/plosone/s/licenses-and-copyright.

7. PLOS ONE now requires that authors provide the original uncropped and unadjusted images underlying all blot or gel results reported in a submission’s figures or Supporting Information files. This policy and the journal’s other requirements for blot/gel reporting and figure preparation are described in detail at https://journals.plos.org/plosone/s/figures#loc-blot-and-gel-reporting-requirements and https://journals.plos.org/plosone/s/figures#loc-preparing-figures-from-image-files. When you submit your revised manuscript, please ensure that your figures adhere fully to these guidelines and provide the original underlying images for all blot or gel data reported in your submission. See the following link for instructions on providing the original image data: https://journals.plos.org/plosone/s/figures#loc-original-images-for-blots-and-gels.

8. Please upload a new copy of Figure 6 as the detail is not clear. Please follow the link for more information: https://journals.plos.org/plosone/s/figures

Reviewers' comments:

Reviewer's Responses to Questions

**Comments to the Author**

1. Is the manuscript technically sound, and do the data support the conclusions?

Reviewer #1: Yes

Reviewer #2: Partly

2. Has the statistical analysis been performed appropriately and rigorously? 

Reviewer #1: Yes

Reviewer #2: I Don't Know

3. Have the authors made all data underlying the findings in their manuscript fully available?

Reviewer #1: Yes

Reviewer #2: Yes

4. Is the manuscript presented in an intelligible fashion and written in standard English?

Reviewer #1: Yes

Reviewer #2: Yes

5. Review Comments to the Author

Reviewer #1: The paper found satisfactory after my review. The author has done all the requisites related to the title mentioned. The results are highly correlated. Research work is highly appreciated. Data produced is genuine.

Reviewer #2: Lack of background depth: The study briefly mentions the global significance of citrus greening but does not explicitly state the current status or prior studies in Nepal. How this study builds upon or differs from existing regional work would strengthen its significance.

Limited sequencing depth: Only five samples were subjected to Sanger sequencing, which is insufficient for strong conclusions on genetic diversity.

No details on PCR validation: The abstract does not mention whether positive and negative controls, sensitivity checks, or replicates were used to confirm PCR reliability.

No environmental or management correlation: The study does not discuss about the factors like altitude, climate, or management practices that might influence disease spread.

The abstract could better link the molecular findings to practical outcomes for farmers or policy.

Future management or diagnostic development prospects are mentioned vaguely without specific pathways.

Time duration to reach the laboratory from collection sites and locations from where the samples have been collected are not mentioned.

6. PLOS authors have the option to publish the peer review history of their article (what does this mean?). If published, this will include your full peer review and any attached files.

Reviewer #1: **Yes:**Dr. Kuljinder Kaur

Reviewer #2: No

---

## [Author Response · Author response to Decision Letter 1]

19 Jan 2026

Detailed review response file uploaded

---

## [Decision Letter · Decision Letter 1]

25 Feb 2026

PONE-D-25-50808R1PCR-Based Detection and Phylogenetic Analysis of Candidatus Liberibacter asiaticus in Citrus Orchards Across NepalPLOS One

Dear Dr. Oliya,

Thank you for submitting your manuscript to PLOS ONE. After careful consideration, we feel that it has merit but does not fully meet PLOS ONE’s publication criteria as it currently stands. Therefore, we invite you to submit a revised version of the manuscript that addresses the points raised during the review process.

We look forward to receiving your revised manuscript.

Kind regards,

Nishant Kumar, Ph.D

Academic Editor

PLOS One

Journal Requirements:

Reviewers' comments:

Reviewer's Responses to Questions

**Comments to the Author**

1. If the authors have adequately addressed your comments raised in a previous round of review and you feel that this manuscript is now acceptable for publication, you may indicate that here to bypass the “Comments to the Author” section, enter your conflict of interest statement in the “Confidential to Editor” section, and submit your "Accept" recommendation.

Reviewer #2: All comments have been addressed

Reviewer #3: All comments have been addressed

2. Is the manuscript technically sound, and do the data support the conclusions?

Reviewer #2: Yes

Reviewer #3: Partly

3. Has the statistical analysis been performed appropriately and rigorously? 

Reviewer #2: Yes

Reviewer #3: Yes

4. Have the authors made all data underlying the findings in their manuscript fully available?

Reviewer #2: Yes

Reviewer #3: Yes

5. Is the manuscript presented in an intelligible fashion and written in standard English?

Reviewer #2: Yes

Reviewer #3: Yes

6. Review Comments to the Author

Reviewer #2: (No Response)

Reviewer #3: Comments to the Author

The author addresses an important phytopathological problem (HLB) and provides country-wide molecular detection data from Nepal, which has regional significance. The dataset 1026 samples is strong, and the study attempts both diagnostics and phylogenetics. However, scientific depth, methodological rigor are insufficient for PLOS ONE in the current form. Several major conceptual and technical limitations must be addressed.

Major comments

Study primarily relies on conventional PCR and 16S rRNA sequencing, which are standard approaches. The claim of being a “comprehensive genetic characterization” is overstated given the limited depth of analysis.

Out of 1026 samples, only five were sequenced without justification of selection criteria. This is insufficient to represent genetic diversity across Nepal.

Phylogeny is based only on a short 16S rRNA fragment (~500 bp), which lacks resolution for intra-species variation. Use of Neighbor-Joining further weakens evolutionary interpretation.

“Candidatus” should be italicized, while “Liberibacter asiaticus” should not be italicized as currently presented. Please ensure consistent and correct formatting of this nomenclature throughout the manuscript.

References are not formatted according to the journal’s guidelines. Please revise all citations and the reference list to strictly comply with the required journal format.

Manuscript lacks essential tables. It is recommended to include tables for primer details, criteria for sample selection for sequencing, and a summary of accession numbers to improve clarity and organization.

Minor Comments

Line 24–25: “Ca. L. asiaticus (Ca. L. asiaticus)” → redundant repetition.

Line 39–40: Sentence is repetitive and can be shortened for clarity.

Line 58: “Gram-negative bacteria Candidatus Liberibacter spp.” → should be “Gram-negative bacterium” or restructure sentence.

Line 208: “25-well thermal cycler” → incorrect terminology; should specify PCR machine model.

Line 209–211: PCR reaction volumes do not sum correctly to 15 µl; needs correction.

Line 370: Missing space before citation “[41]”

Line 323–324: “citrus tresteza virus” → spelling error (tristeza).

Conclusion section: Repeats statements from abstract; should be more concise and focused on implications.

Please Cite these below references

Bharsakale, R. D., Alex, B. K., Gubyad, M. G., Kokane, A. D., Kokane, S. B., Shukla, P. K., & Ghosh, D. K. (2025). Development of a novel duplex PCR-lateral flow immunoassay (d-PCR-LFIA) for simultaneous detection of Candidatus Liberibacter asiaticus and citrus tristeza virus. Tropical Plant Pathology, 50(1), 77.

Ghosh, D., Kokane, S., Savita, B. K., Kumar, P., Sharma, A. K., Ozcan, A., ... & Santra, S. (2022). Huanglongbing pandemic: current challenges and emerging management strategies. Plants, 12(1), 160.

For the listed above comments, I would recommend to the editor accept with major revisions for this manuscript.

7. PLOS authors have the option to publish the peer review history of their article (what does this mean?). If published, this will include your full peer review and any attached files.

Reviewer #2: No

Reviewer #3: **Yes:**Dilip Kumar Ghosh

---

## [Author Response · Author response to Decision Letter 2]

11 Apr 2026

I have attached my responses to reviewer comments as a file named 'Response to Reviewers'.

---

## [Decision Letter · Decision Letter 2]

3 May 2026

PCR-Based Detection and Phylogenetic Analysis of Candidatus Liberibacter asiaticus in Citrus Orchards Across Nepal

PONE-D-25-50808R2

**Dear Dr. Oliya,**

We’re pleased to inform you that your manuscript has been judged scientifically suitable for publication and will be formally accepted for publication once it meets all outstanding technical requirements.

Kind regards,

Nishant Kumar, Ph.D

Academic Editor

PLOS One

Additional Editor Comments (optional):

-

Reviewers' comments:

Reviewer's Responses to Questions

**Comments to the Author**

1. If the authors have adequately addressed your comments raised in a previous round of review and you feel that this manuscript is now acceptable for publication, you may indicate that here to bypass the “Comments to the Author” section, enter your conflict of interest statement in the “Confidential to Editor” section, and submit your "Accept" recommendation.

Reviewer #3: All comments have been addressed

Reviewer #4: All comments have been addressed

2. Is the manuscript technically sound, and do the data support the conclusions?

Reviewer #3: Yes

Reviewer #4: Yes

3. Has the statistical analysis been performed appropriately and rigorously? 

Reviewer #3: Yes

Reviewer #4: Yes

4. Have the authors made all data underlying the findings in their manuscript fully available?

Reviewer #3: Yes

Reviewer #4: Yes

5. Is the manuscript presented in an intelligible fashion and written in standard English?

Reviewer #3: Yes

Reviewer #4: Yes

6. Review Comments to the Author

Reviewer #3: Authors have addressed all the major and minor comments raised during the previous rounds of review. The manuscript has been substantially improved in terms of clarity, methodological transparency, and scientific accuracy. All previously identified concerns have been satisfactorily resolved, and the manuscript is now scientifically sound and suitable for publication.

Reviewer #4: The manuscript is well-structured, clearly written, and presents scientifically sound work. The objectives are clearly defined, and the methodology is appropriate and adequately described, allowing reproducibility. The data are presented in a coherent manner and strongly support the conclusions drawn by the authors. The statistical analyses are robust and properly interpreted. The discussion effectively contextualizes the findings within the existing literature, highlighting the significance and novelty of the work. Overall, this manuscript makes a valuable contribution to the field and is suitable for publication in its current form.

7. PLOS authors have the option to publish the peer review history of their article (what does this mean?). If published, this will include your full peer review and any attached files.

Reviewer #3: No

Reviewer #4: No

---

## [Editor Report · Acceptance letter]

PONE-D-25-50808R2

PLOS One

Dear Dr. Oliya,

I'm pleased to inform you that your manuscript has been deemed suitable for publication in PLOS One. Congratulations! Your manuscript is now being handed over to our production team.

Kind regards,

on behalf of

Dr. Nishant Kumar

Academic Editor

PLOS One